# Fader Networks:
# Manipulating Images by Sliding Attributes

**Guillaume Lample**[1,2], **Neil Zeghidour**[1,3], **Nicolas Usunier**[1],
**Antoine Bordes**[1], **Ludovic Denoyer**[2], **Marc'Aurelio Ranzato**[1]
{gl,neilz,usunier,abordes,ranzato}@fb.com
ludovic.denoyer@lip6.fr

## Abstract

This paper introduces a new encoder-decoder architecture that is trained to re-construct images by disentangling the salient information of the image and the values of attributes directly in the latent space. As a result, after training, our model can generate different realistic versions of an input image by varying the attribute values. By using continuous attribute values, we can choose how much a specific attribute is perceivable in the generated image. This property could allow for applications where users can modify an image using sliding knobs, like *faders* on a mixing console, to change the facial expression of a portrait, or to update the color of some objects. Compared to the state-of-the-art which mostly relies on training adversarial networks in pixel space by altering attribute values at train time, our approach results in much simpler training schemes and nicely scales to multiple attributes. We present evidence that our model can significantly change the perceived value of the attributes while preserving the naturalness of images.

## 1 Introduction

We are interested in the problem of manipulating natural images by controlling some attributes of interest. For example, given a photograph of the face of a person described by their gender, age, and expression, we want to generate a realistic version of this same person looking older or happier, or an image of a hypothetical twin of the opposite gender. This task and the related problem of unsupervised domain transfer recently received a lot of interest [18, 25, 10, 27, 22, 24], as a case study for conditional generative models but also for applications like automatic image edition. The key challenge is that the transformations are ill-defined and training is unsupervised: the training set contains images annotated with the attributes of interest, but there is no example of the transformation: In many cases such as the "gender swapping" example above, there are no pairs of images representing the same person as a male or as a female. In other cases, collecting examples requires a costly annotation process, like taking pictures of the same person with and without glasses.

Our approach relies on an encoder-decoder architecture where, given an input image $x$ with its attributes $y$, the encoder maps $x$ to a latent representation $z$, and the decoder is trained to reconstruct $x$ given $(z, y)$. At inference time, a test image is encoded in the latent space, and the user chooses the attribute values $y$ that are fed to the decoder. Even with binary attribute values at train time, each attribute can be considered as a continuous variable during inference to control how much it is perceived in the final image. We call our architecture *Fader Networks*, in analogy to the sliders of an audio mixing console, since the user can choose how much of each attribute they want to incorporate.

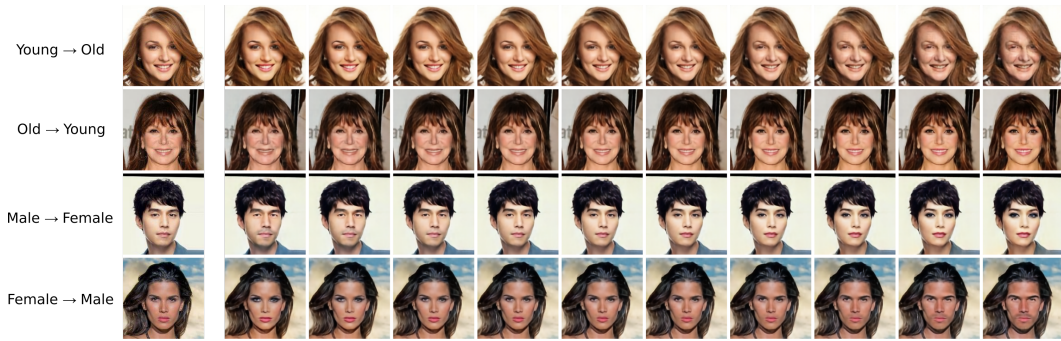

Figure 1: Interpolation between different attributes (Zoom in for better resolution). Each line shows reconstructions of the same face with different attribute values, where each attribute is controlled as a continuous variable. It is then possible to make an old person look older or younger, a man look more manly or to imagine his female version. Left images are the originals.

The fundamental feature of our approach is to constrain the latent space to be invariant to the attributes of interest. Concretely, it means that the distribution over images of the latent representations should be identical for all possible attribute values. This invariance is obtained by using a procedure similar to domain-adversarial training (see e.g., [21, 6, 15]). In this process, a classifier learns to predict the attributes $y$ given the latent representation $z$ during training while the encoder-decoder is trained based on two objectives at the same time. The first objective is the reconstruction error of the decoder, i.e., the latent representation $z$ must contain enough information to allow for the reconstruction of the input. The second objective consists in fooling the attribute classifier, i.e., the latent representation must prevent it from predicting the correct attribute values. In this model, achieving invariance is a means to filter out, or hide, the properties of the image that are related to the attributes of interest. A single latent representation thus corresponds to different images that share a common structure but with different attribute values. The reconstruction objective then forces the decoder to use the attribute values to choose, from the latent representation, the intended image.

Our motivation is to learn a disentangled latent space in which we have explicit control on some attributes of interest, without supervision of the intended result of modifying attribute values. With a similar motivation, several approaches have been tested on the same tasks [18, 25], on related image-to-image translation problems [10, 27], or for more specific applications like the creation of parametrized avatars [24]. In addition to a reconstruction loss, the vast majority of these works rely on adversarial training in pixel space, which compares during training images generated with an intentional change of attributes from genuine images for the target attribute values. Our approach is different both because we use adversarial training for the latent space instead of the output, but also because adversarial training aims at learning invariance to attributes. The assumption underlying our work is that a high fidelity to the input image is less conflicting with the invariance criterion, than with a criterion that forces the hallucinated image to match images from the training set.

As a consequence of this principle, our approach results in much simpler training pipelines than those based on adversarial training in pixel space, and is readily amenable to controlling multiple attributes, by adding new output variables to the discriminator of the latent space. As shown in Figure 1 on test images from the CelebA dataset [14], our model can make subtle changes to portraits that end up sufficient to alter the perceived value of attributes while preserving the natural aspect of the image and the identity of the person. Our experiments show that our model outperforms previous methods based on adversarial training on the decoders' output like [18] in terms of both reconstruction loss and generation quality as measured by human subjects. We believe this disentanglement approach is a serious competitor to the widespread adversarial losses on the decoder output for such tasks.

In the remainder of the paper, we discuss in more details the related work in Section 2. We then present the training procedure in Section 3 before describing the network architecture and the implementation in Section 4. Experimental results are shown in Section 5.

## 2 Related work

There is substantial literature on attribute-based and/or conditional image generation that can be split in terms of required supervision, with three different levels. At one extreme are fully supervised approaches developed to model known transformations, where examples take the form of *(input, transformation, result of the transformation)*. In that case, the model needs to learn the desired transformation. This setting was previously explored to learn affine transformations [9], 3D rotations [26], lighting variations [12] and 2D video game animations [20]. The methods developed in these works however rely on the supervised setting, and thus cannot be applied in our setup.

At the other extreme of the supervision spectrum lie fully unsupervised methods that aim at learning deep neural networks that disentangle the factors of variations in the data, without specification of the attributes. Example methods are InfoGAN [4], or the predictability minimization framework proposed in [21]. The neural photo editor [3] disentangles factors of variations in natural images for image edition. [8] introduced the beta-VAE, a modification of the variational autoencoder (VAE) framework that can learn latent factorized representations in a completely unsupervised manner. This setting is considerably harder than the one we consider, and in general, it may be difficult with these methods to automatically discover high-level concepts such as gender or age.

Our work lies in between the two previous settings. It is related to information as in [16]. Methods developed for unsupervised domain transfer [10, 27, 22, 24] can also be applied in our case: given two different domains of images such as "drawings" and "photograph", one wants to map an image from one domain to the other without supervision; in our case, a domain would correspond to an attribute value. The mappings are trained using adversarial training in pixel space as mentioned in the introduction, using separate encoders and/or decoders per domain, and thus do not scale well to multiple attributes. In this line of work but more specifically considering the problem of modifying attributes, the Invertible conditional GAN [18] first trains a GAN conditioned on the attribute values, and in a second step learns to map input images to the latent space of the GAN, hence the name of invertible GANs. It is used as a baseline in our experiments. Antipov et al. [1] use a pre-trained face recognition system instead of a conditional GAN to learn the latent space, and only focuses on the age attribute. The attribute-to-image approach [25] is a variational auto-encoder that disentangles foreground and background to generate images using attribute values only. Conditional generation is performed by inferring the latent state given the correct attributes and then changing the attributes.

Additionally, our work is related to work on learning invariant latent spaces using adversarial training in domain adaptation [6], fair classification [5] and robust inference [15]. The training criterion we use for enforcing invariance is similar to the one used in those works, the difference is that the end-goal of these works is only to filter out nuisance variables or sensitive information. In our case, we learn generative models, and invariance is used as a means to force the decoder to use attribute information in its reconstruction.

Finally, for the application of automatically modifying faces using attributes, the feature interpolation approach of [23] presents a means to generate alterations of images based on attributes using a pre-trained network on ImageNet. While their approach is interesting from an application perspective, their inference is costly and since it relies on pre-trained models, cannot naturally incorporate factors or attributes that have not been foreseen during the pre-training.

## 3 Fader Networks

Let $\mathcal{X}$ be an image domain and $\mathcal{Y}$ the set of possible attributes associated with images in $\mathcal{X}$, where in the case of people's faces typical attributes are *glasses/no glasses*, *man/woman*, *young/old*. For simplicity, we consider here the case where attributes are binary, but our approach could be extended to categorical attributes. In that setting, $\mathcal{Y} = \{0, 1\}^n$, where $n$ is the number of attributes. We have a training set $\mathcal{D} = \{(x^1, y^1), ..., (x^m, y^m)\}$, of $m$ pairs (image, attribute) $(x^i \in \mathcal{X}, y^i \in \mathcal{Y})$. The end goal is to learn from $\mathcal{D}$ a model that will generate, for any attribute vector $y'$, a version of an input image $x$ whose attribute values correspond to $y'$.

**Encoder-decoder architecture**    Our model, described in Figure 2, is based on an encoder-decoder architecture with domain-adversarial training on the latent space. The encoder $E_{\theta_{\mathrm{enc}}} : \mathcal{X} \to \mathbb{R}^N$ is a convolutional neural network with parameters $\theta_{\mathrm{enc}}$ that maps an input image to its $N$-dimensional latent representation $E_{\theta_{\mathrm{enc}}}(x)$. The decoder $D_{\theta_{\mathrm{dec}}} : (\mathbb{R}^N, \mathcal{Y}) \to \mathcal{X}$ is a deconvolutional network with parameters $\theta_{\mathrm{dec}}$ that produces a new version of the input image given its latent representation $E_{\theta_{\mathrm{enc}}}(x)$

and any attribute vector $y'$. When the context is clear, we simply use $D$ and $E$ to denote $D_{\theta_\text{dec}}$ and $E_{\theta_\text{enc}}$. The precise architectures of the neural networks are described in Section 4. The auto-encoding loss associated to this architecture is a classical mean squared error (MSE) that measures the quality of the reconstruction of a training input $x$ given its true attribute vector $y$:

$$\mathcal{L}_\text{AE}(\theta_\text{enc}, \theta_\text{dec}) = \frac{1}{m} \sum_{(x,y) \in \mathcal{D}} \left\| D_{\theta_\text{dec}}\big(E_{\theta_\text{enc}}(x), y\big) - x \right\|_2^2$$

The exact choice of the reconstruction loss is not fundamental in our approach, and adversarial losses such as PatchGAN [13] could be used in addition to the MSE at this stage to obtain better textures or sharper images, as in [10]. Using a mean absolute or mean squared error is still necessary to ensure that the reconstruction matches the original image.

Ideally, modifying $y$ in $D(E(x), y)$ would generate images with different perceived attributes, but similar to $x$ in every other aspect. However, without additional constraints, the decoder learns to ignore the attributes, and modifying $y$ at test time has no effect.

**Learning attribute-invariant latent representations**    To avoid this behavior, our approach is to learn latent representations that are invariant with respect to the attributes. By invariance, we mean that given two versions of a same object $x$ and $x'$ that are the same up to their attribute values, for instance two images of the same person with and without glasses, the two latent representations $E(x)$ and $E(x')$ should be the same. When such an invariance is satisfied, the decoder must use the attribute to reconstruct the original image. Since the training set does not contain different versions of the same image, this constraint cannot be trivially added in the loss.

We hence propose to incorporate this constraint by doing adversarial training on the latent space. This idea is inspired by the work on predictability minimization [21] and adversarial training for domain adaptation [6, 15] where the objective is also to learn an invariant latent representation using an adversarial formulation of the learning objective. To that end, an additional neural network called the *discriminator* is trained to identify the true attributes $y$ of a training pair $(x, y)$ given $E(x)$. The invariance is obtained by learning the encoder $E$ such that the discriminator is unable to identify the right attributes. As in GANs [7], this corresponds to a two-player game where the discriminator aims at maximizing its ability to identify attributes, and $E$ aims at preventing it to be a good discriminator. The exact structure of our discriminator is described in Section 4.

**Discriminator objective**    The discriminator outputs probabilities of an attribute vector $P_{\theta_\text{dis}}(y|E(x))$, where $\theta_\text{dis}$ are the discriminator's parameters. Using the subscript $k$ to refer to the $k$-th attribute, we have $\log P_{\theta_\text{dis}}(y|E(x)) = \sum\limits_{k=1}^{n} \log P_{\theta_\text{dis},k}(y_k|E(x))$. Since the objective of the discriminator is to predict the attributes of the input image given its latent representation, its loss depends on the current state of the encoder and is written as:

$$\mathcal{L}_\text{dis}(\theta_\text{dis}|\theta_\text{enc}) = -\frac{1}{m} \sum_{(x,y) \in \mathcal{D}} \log P_{\theta_\text{dis}}\big(y \big| E_{\theta_\text{enc}}(x)\big) \tag{1}$$

**Adversarial objective**    The objective of the encoder is now to compute a latent representation that optimizes two objectives. First, the decoder should be able to reconstruct $x$ given $E(x)$ and $y$, and at the same time the discriminator should not be able to predict $y$ given $E(x)$. We consider that a mistake is made when the discriminator predicts $1 - y_k$ for attribute $k$. Given the discriminator's parameters, the complete loss of the encoder-decoder architecture is then:

$$\mathcal{L}(\theta_\text{enc}, \theta_\text{dec}|\theta_\text{dis}) = \frac{1}{m} \sum_{(x,y) \in \mathcal{D}} \left\| D_{\theta_\text{dec}}\big(E_{\theta_\text{enc}}(x), y\big) - x \right\|_2^2 - \lambda_E \log P_{\theta_\text{dis}}(1 - y | E_{\theta_\text{enc}}(x)), \tag{2}$$

where $\lambda_E > 0$ controls the trade-off between the quality of the reconstruction and the invariance of the latent representations. Large values of $\lambda_E$ will restrain the amount of information about $x$ contained in $E(x)$, and result in blurry images, while low values limit the decoder's dependency on the latent code $y$ and will result in poor effects when altering attributes.

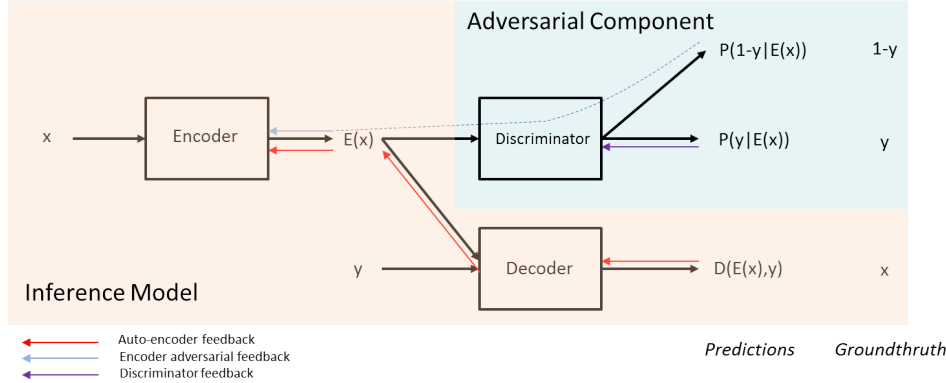

Figure 2: Main architecture. An (image, attribute) pair $(x, y)$ is given as input. The encoder maps $x$ to the latent representation $z$; the discriminator is trained to predict $y$ given $z$ whereas the encoder is trained to make it impossible for the discriminator to predict $y$ given $z$ only. The decoder should reconstruct $x$ given $(z, y)$. At test time, the discriminator is discarded and the model can generate different versions of $x$ when fed with different attribute values.

**Learning algorithm**   Overall, given the current state of the encoder, the optimal discriminator parameters satisfy $\theta^*_{\mathrm{dis}}(\theta_{\mathrm{enc}}) \in \mathrm{argmin}_{\theta_{\mathrm{dis}}} \mathcal{L}_{\mathrm{dis}}(\theta_{\mathrm{dis}}|\theta_{\mathrm{enc}})$. If we ignore problems related to multiple (and local) minima, the overall objective function is

$$\theta^*_{\mathrm{enc}}, \theta^*_{\mathrm{dec}} = \underset{\theta_{\mathrm{enc}}, \theta_{\mathrm{dec}}}{\mathrm{argmin}}\, \mathcal{L}(\theta_{\mathrm{enc}}, \theta_{\mathrm{dec}}|\theta^*_{\mathrm{dis}}(\theta_{\mathrm{enc}}))\,.$$

In practice, it is unreasonable to solve for $\theta^*_{\mathrm{dis}}(\theta_{\mathrm{enc}})$ at each update of $\theta_{\mathrm{enc}}$. Following the practice of adversarial training for deep networks, we use stochastic gradient updates for all parameters, considering the current value of $\theta_{\mathrm{dis}}$ as an approximation for $\theta^*_{\mathrm{dis}}(\theta_{\mathrm{enc}})$. Given a training example $(x, y)$, let us denote $\mathcal{L}_{\mathrm{dis}}(\theta_{\mathrm{dis}}|\theta_{\mathrm{enc}}, x, y)$ the auto-encoder loss restricted to $(x, y)$ and $\mathcal{L}(\theta_{\mathrm{enc}}, \theta_{\mathrm{dec}}|\theta_{\mathrm{dis}}, x, y)$ the corresponding discriminator loss. The update at time $t$ given the current parameters $\theta^{(t)}_{\mathrm{dis}}$, $\theta^{(t)}_{\mathrm{enc}}$, and $\theta^{(t)}_{\mathrm{dec}}$ and the training example $(x^{(t)}, y^{(t)})$ is:

$$\theta^{(t+1)}_{\mathrm{dis}} = \theta^{(t)}_{\mathrm{dis}} - \eta \nabla_{\theta_{\mathrm{dis}}} \mathcal{L}_{\mathrm{dis}}\big(\theta^{(t)}_{\mathrm{dis}}|\theta^{(t)}_{\mathrm{enc}}, x^{(t)}, y^{(t)}\big)$$
$$[\theta^{(t+1)}_{\mathrm{enc}}, \theta^{(t+1)}_{\mathrm{dec}}] = [\theta^{(t)}_{\mathrm{enc}}, \theta^{(t)}_{\mathrm{dec}}] - \eta \nabla_{\theta_{\mathrm{enc}}, \theta_{\mathrm{dec}}} \mathcal{L}\big(\theta^{(t)}_{\mathrm{enc}}, \theta^{(t)}_{\mathrm{dec}}|\theta^{(t+1)}_{\mathrm{dis}}, x^{(t)}, y^{(t)}\big)\,.$$

The details of training and models are given in the next section.

## 4   Implementation

We adapt the architecture of our network from [10]. Let $C_k$ be a Convolution-BatchNorm-ReLU layer with $k$ filters. Convolutions use kernel of size $4 \times 4$, with a stride of 2, and a padding of 1, so that each layer of the encoder divides the size of its input by 2. We use leaky-ReLUs with a slope of 0.2 in the encoder, and simple ReLUs in the decoder.

The encoder consists of the following 7 layers:

$$C_{16} - C_{32} - C_{64} - C_{128} - C_{256} - C_{512} - C_{512}$$

Input images have a size of $256 \times 256$. As a result, the latent representation of an image consists of 512 feature maps of size $2 \times 2$. In our experiments, using 6 layers gave us similar results, while 8 layers significantly decreased the performance, even when using more feature maps in the latent state.

To provide the decoder with image attributes, we append the latent code to each layer given as input to the decoder, where the latent code of an image is the concatenation of the one-hot vectors representing

| Model | Naturalness | | | Accuracy | | |
|---|---|---|---|---|---|---|
| | Mouth | Smile | Glasses | Mouth | Smile | Glasses |
| Real Image | 92.6 | 87.0 | 88.6 | 89.0 | 88.3 | 97.6 |
| IcGAN AE | 22.7 | 21.7 | 14.8 | 88.1 | 91.7 | 86.2 |
| IcGAN Swap | 11.4 | 22.9 | 9.6 | 10.1 | 9.9 | 47.5 |
| FadNet AE | 88.4 | 75.2 | 78.8 | 91.8 | 90.1 | 94.5 |
| FadNet Swap | 79.0 | 31.4 | 45.3 | 66.2 | 97.1 | 76.6 |

Table 1: Perceptual evaluation of naturalness and swap accuracy for each model. The naturalness score is the percentage of images that were labeled as "real" by human evaluators to the question "Is this image a real photograph or a fake generated by a graphics engine?". The accuracy score is the classification accuracy by human evaluators on the values of each attribute.

the values of its attributes (binary attributes are represented as $[1, 0]$ and $[0, 1]$). We append the latent code as additional constant input channels for all the convolutions of the decoder. Denoting by $n$ the number of attributes, (hence a code of size $2n$), the decoder is symmetric to the encoder, but uses transposed convolutions for the up-sampling:

$$C_{512+2n} - C_{512+2n} - C_{256+2n} - C_{128+2n} - C_{64+2n} - C_{32+2n} - C_{16+2n} \ .$$

The discriminator is a $C_{512}$ layer followed by a fully-connected neural network of two layers of size $512$ and $n$ repsectively.

**Dropout** We found it beneficial to add dropout in our discriminator. We hypothesized that dropout helped the discriminator to rely on a wider set of features in order to infer the current attributes, improving and stabilizing its accuracy, and consequently giving better feedback to the encoder. We set the dropout rate to $0.3$ in all our experiments. Following [10], we also tried to add dropout in the first layers of the decoder, but in our experiments, this turned out to significantly decrease the performance.

**Discriminator cost scheduling** Similarly to [2], we use a variable weight for the discriminator loss coefficient $\lambda_E$. We initially set $\lambda_E$ to 0 and the model is trained like a normal auto-encoder. Then, $\lambda_E$ is linearly increased to $0.0001$ over the first $500,000$ iterations to slowly encourage the model to produce invariant representations. This scheduling turned out to be critical in our experiments. Without it, we observed that the encoder was too affected by the loss coming from the discriminator, even for low values of $\lambda_E$.

**Model selection** Model selection was first performed automatically using two criteria. First, we used the reconstruction error on original images as measured by the MSE. Second, we also want the model to properly swap the attributes of an image. For this second criterion, we train a classifier to predict image attributes. At the end of each epoch, we swap the attributes of each image in the validation set and measure how well the classifier performs on the decoded images. These two metrics were used to filter out potentially good models. The final model was selected based on human evaluation on images from the train set reconstructed with swapped attributes.

## 5 Experiments

### 5.1 Experiments on the celebA dataset

**Experimental setup** We first present experiments on the celebA dataset [14], which contains $200,000$ images of celebrity of shape $178 \times 218$ annotated with $40$ attributes. We used the standard training, validation and test split. All pictures presented in the paper or used for evaluation have been taken from the test set. For pre-processing, we cropped images to $178 \times 178$, and resized them to $256 \times 256$, which is the resolution used in all figures of the paper. Image values were normalized to $[-1, 1]$. All models were trained with Adam [11], using a learning rate of $0.002$, $\beta_1 = 0.5$, and a batch size of 32. We performed data augmentation by flipping horizontally images with a probability $0.5$ at each iteration. As model baseline, we used IcGAN [18] with the model provided by the authors and trained on the same dataset. [4]

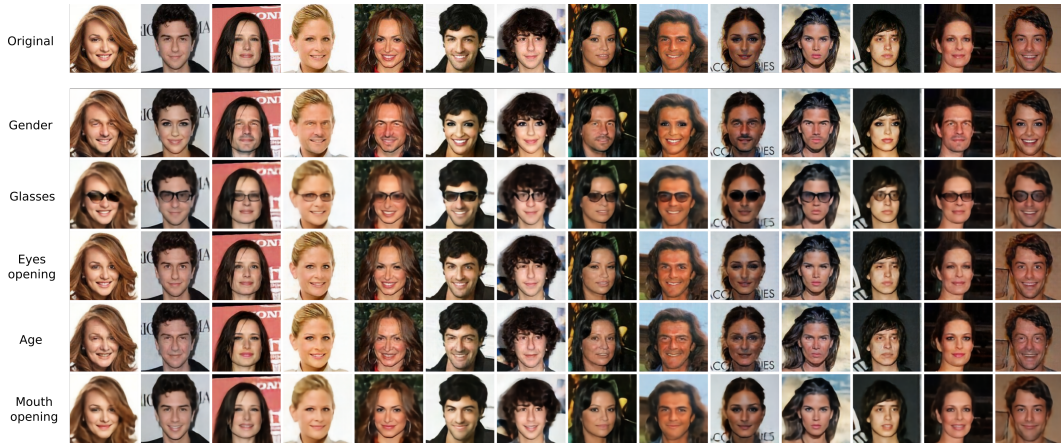

Figure 3: Swapping the attributes of different faces. Zoom in for better resolution.

**Qualitative evaluation**   Figure 3 shows examples of images generated when swapping different attributes: the generated images have a high visual quality and clearly handle the attribute value changes, for example by adding realistic glasses to the different faces. These generated images confirm that the latent representation learned by Fader Networks is both invariant to the attribute values, but also captures the information needed to generate any version of a face, for any attribute value. Indeed, when looking at the shape of the generated glasses, different glasses shapes and colors have been integrated into the original face depending on the face: our model is not only adding "generic" glasses to all faces, but generates plausible glasses depending on the input.

**Quantitative evaluation protocol**   We performed a quantitative evaluation of Fader Networks on Mechanical Turk, using IcGAN as a baseline. We chose the three attributes *Mouth (Open/Close)*, *Smile (With/Without)* and *Glasses (With/Without)* as they were attributes in common between IcGAN and our model. We evaluated two different aspects of the generated images: the **naturalness**, that measures the quality of generated images, and the **accuracy**, that measures how well swapping an attribute value is reflected in the generation. Both measures are necessary to assess that we generate natural images, and that the swap is effective. We compare: REAL IMAGE , that provides original images without transformation, FADNET AE and ICGAN AE , that reconstruct original images without attribute alteration, and FADNET SWAP and ICGAN SWAP , that generate images with one swapped attribute, e.g., *With Glasses → Without Glasses*. Before being submitted to Mechanical Turk, all images were cropped and resized following the same processing than IcGAN. As a result, output images were displayed in $64 \times 64$ resolution, also preventing Workers from basing their judgment on the sharpness of presented images exclusively.

Technically, we should also assess that the identity of a person is preserved when swapping attributes. This seemed to be a problem for GAN-based methods, but the reconstruction quality of our model is very good (RMSE on test of $0.0009$, to be compared to $0.028$ for IcGAN), and we did not observe this issue. Therefore, we did not evaluate this aspect.

For naturalness, the first $500$ images from the test set such that there are $250$ images for each attribute value were shown to Mechanical Turk Workers, $100$ for each of the $5$ different models presented above. For each image, we asked whether the image seems natural or generated. The description given to the Workers to understand their task showed $4$ examples of real images, and $4$ examples of fake images (1 FADNET AE , 1 FADNET SWAP , 1 ICGAN AE , 1 ICGAN SWAP ).

The accuracy of each model on each attribute was evaluated in a different classification task, resulting in a total of 15 experiments. For example, the FadNet/Glasses experiment consisted in asking Workers whether people with glasses being added by FADNET SWAP effectively possess glasses, and vice-versa. This allows us to evaluate how perceptible the swaps are to the human eye. In each experiment, $100$ images were shown ($50$ images per class, in the order they appear in the test set).

In both quantitative evaluations, each experiment was performed by $10$ Workers, resulting in $5,000$ samples per experiment for naturalness, and $1,000$ samples per classification experiment on swapped attributes. The results on both tasks are shown in Table 1.

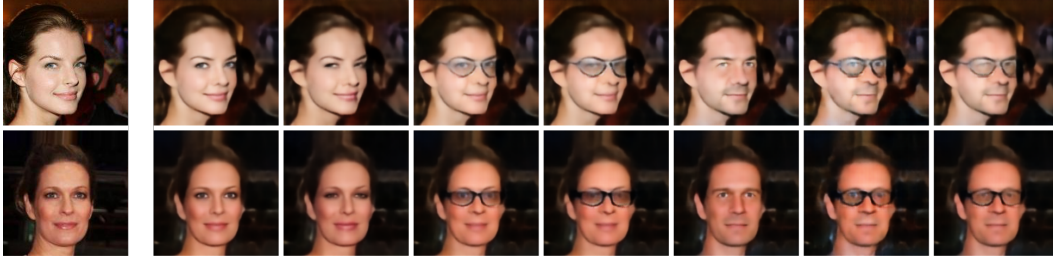

Figure 4: (Zoom in for better resolution.) Examples of multi-attribute swap (Gender / Opened eyes / Eye glasses) performed by the same model. Left images are the originals.

**Quantitative results**  In the naturalness experiments, only around $90\%$ of real images were classified as "real" by the Workers, indicating the high level of requirement to generate natural images. Our model obtained high naturalness accuracies when reconstructing images without swapping attributes: $88.4\%$, $75.2\%$ and $78.8\%$, compared to IcGAN reconstructions whose accuracy does not exceed $23\%$, whether it be for reconstructed or swapped images. For the swap, FADNET SWAP still consistently outperforms ICGAN SWAP by a large margin. However, the naturalness accuracy varies a lot based on the swapped attribute: from $79.0\%$ for the opening of the mouth, down to $31.4\%$ for the smile.

Classification experiments show that reconstructions with FADNET AE and ICGAN AE have very high classification scores, and are even on par with real images on both Mouth and Smile. FADNET SWAP obtains an accuracy of $66.2\%$ for the mouth, $76.6\%$ for the glasses and $97.1\%$ for the smile, indicating that our model can swap these attributes with a very high efficiency. On the other hand, with accuracies of $10.1\%$, $47.5\%$ and $9.9\%$ on these same attributes, ICGAN SWAP does not seem able to generate convincing swaps.

**Multi-attributes swapping**  We present qualitative results for the ability of our model to swap multiple attributes at once in Figure 4, by jointly modifying the gender, open eyes and glasses attributes. Even in this more difficult setting, our model can generate convincing images with multiple swaps.

## 5.2  Experiments on Flowers dataset

We performed additional experiments on the Oxford-102 dataset, which contains about $9,000$ images of flowers classified into $102$ categories [17]. Since the dataset does not contain other labels than the flower categories, we built a list of color attributes from the flower captions provided by [19]. Each flower is provided with 10 different captions. For a given color, we gave a flower the associated color attribute, if that color appears in at least 5 out of the 10 different captions. Although being naive, this approach was enough to create accurate labels. We resized images to $64 \times 64$. Figure 5 represents reconstructed flowers with different values of the "pink" attribute. We can observe that the color of the flower changes in the desired direction, while keeping the background cleanly unchanged.

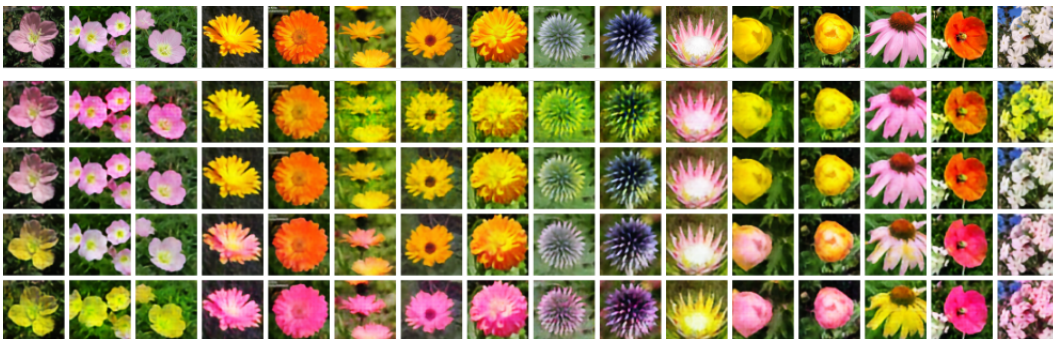

Figure 5: Examples of reconstructed flowers with different values of the *pink* attribute. First row images are the originals. Increasing the value of that attribute will turn flower colors into pink, while decreasing it in images with originally pink flowers will make them turn yellow or orange.

# 6 Conclusion

We presented a new approach to generate variations of images by changing attribute values. The approach is based on enforcing the invariance of the latent space w.r.t. the attributes. A key advantage of our method compared to many recent models [27, 10] is that it generates realistic images of high resolution without needing to apply a GAN to the decoder output. As a result, it could easily be extended to other domains like speech, or text, where the backpropagation through the decoder can be really challenging because of the non-differentiable text generation process for instance. However, methods commonly used in vision to assess the visual quality of the generated images, like PatchGAN, could totally be applied on top of our model.

### Acknowledgments

The authors would like to thank Yedid Hoshen for initial discussions about the core ideas of the paper, Christian Pursch and Alexander Miller for their help in setting up the experiments and Mechanical Turk evaluations. The authors are also grateful to David Lopez-Paz and Mouhamadou Moustapha Cisse for useful feedback and support on this project.

## Footnotes

[1]Facebook AI Research

[2]Sorbonne Universités, UPMC Univ Paris 06, UMR 7606, LIP6

[3]LSCP, ENS, EHESS, CNRS, PSL Research University, INRIA

[4] `https://github.com/Guim3/IcGAN`

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
