[Reviews · NeurIPS 2017]

Reviewer 1



The authors propose a way to learn a disentangled representation of images that scales to reasonably large inputs (256x256), and that can be used to generate natural looking variations on these images by traversing the learnt attributes. The authors do so by using labelled attributes information for supervision and augmenting an autoencoder setup with an additional GAN objective in the latent space (the GAN discriminator is trying to guess the attributes from the latent space of the autoencoder). The proposed Fader networks perform well in terms of learning a disentangled representation and generating natural looking sampled with swapped attributes. However, its performance is not perfect (as indicated by the human evaluation scores) and the overall novelty of the approach seems to be somewhat incremental. Saying that, the paper is very well written. One thing I found surprising was that the authors did not discuss beta-VAE (Higgins et al, 2017) in their literature review. When discussing unsupervised approaches to disentangled factor learning the authors mention that the methods they cite find difficult to "automatically discover high-level concepts such as gender or age". Beta-VAE was able to discover age and gender as factors of variation on celebA in a completely unsupervised manner. Of course, being a VAE based method it did suffer from blurry samples, but I feel that the authors should cite the paper given that it achieves their stated objective of learning a "fader"-like architecture.

Reviewer 2



The paper claims to introduce a new encoder/decoder that can disentangle meaningful attributes (eg. young vs old, male vs female) in the latent representation. These disentangled attributes can be "faded" up or down to generate images that vary in how strongly the attribute is expressed. The problem they tackle is unsupervised learning of images, where the images come with semantic attributes, but the data does not specify transformations between them. The core idea is to learn to encode an image in a latent space in a way that is agnostic to the attributes of interest. In effect, it learns to infer a posterior over an image encoding with the attributes of interest marginalized out. This way, the attribute can be input at generation time to effectively apply the constraint that only images with that attribute should be generated. They acknowledge this is related to domain-adversarial training. Their idea is implemented via adversarial training on the latent representations. A "discriminator" network is trained to predict the input attribute from the latent encoding, and the encoder's training is penalized in proportion to the discriminator's performance. There is also a reconstruction objective for the decoder. The architecture uses a standard mixture of CNNs, relus, batchnorm, and dropout. To generate images with different attributes, the target attribute is simply input at test time. This allows generated images to vary between attributes such as age, gender, glasses, eye opening, and mouth opening. The results are quite convincing. The generated images are realistic, and for some attributes (eg. gender) the transformations are very good. Some are a little weaker, but those can probably be remedied with some tweaking. Overall the work is solid and merits publication. The contribution isn't transformative, but the idea is simple, effective and generally important. Everything was executed well in the paper and the writing is clear and easy to follow.

Reviewer 3



*** Summary *** This paper aims to augment an autoencoder with the ability to tweak certain (known) attributes. These attributes are known at training time and for a dataset of faces include aspects like [old vs young], [smiling vs not smiling], etc. They hope to be able to tweak these attributes along a continuous spectrum, even when the labels only occur as binary values. To achieve this they propose an (encoder, decoder) setup where the encoder maps the image x to a latent vector z and then the decoder produces an image taking z, together with the attributes y as inputs. When such a network is trained in the ordinary fashion, the decoder learns to ignore y because z already encodes everything that the network needs to know. To compel the decoder network to use y, the authors propose introducing a adversarial learning framework in which a discriminator D is trained to infer the attributes from z. Thus the encoder must produce representations that are invariant to the attributes y. The writing is clear and any strong researcher should be able to reproduce their results from the presentation here. *** Conclusions *** This paper presents a nice idea. Not an inspiring, or world changing idea, but a worthy enough idea. Moreover, this technique offers a solution that large number of people are actually interested in (for good or for malice). What makes this paper a clear accept is that the authors take this simple useful idea and execute it wonderfully. The presentation is crystal clear. The experiments are compelling (both quantitatively and qualitatively). The live study with Mechanical Turk workers answers some key questions any reasonable reviewer might ask. I expect this work to actually be used in the real world and happily champion it for acceptance at NIPS 2017. *** Small Writing Quibble *** "We found it extremely beneficial to add dropout" Adverbs of degree (like "very", "extremely", etc) add nothing of value to the sentence. Moreover, it expresses an opinion in a sly fashion. Instead of saying dropout is "extremely" important, say ***why*** it is important. What happens when you don't use dropout? How do the results compare qualitatively and quantitatively?